# Clinical Efficacy and Safety of Antimicrobial Photodynamic Therapy in Residual Periodontal Pockets during the Maintenance Phase

**DOI:** 10.3390/ph15080924

**Published:** 2022-07-25

**Authors:** Yasunori Yamashita, Megumi Mae, Masayuki Oohira, Yukio Ozaki, Seigo Ohba, Izumi Asahina, Atsutoshi Yoshimura

**Affiliations:** 1Department of Periodontology and Endodontology, Nagasaki University Graduate School of Biomedical Sciences, 1-7-1 Sakamoto, Nagasaki 852-8588, Japan; m.mae@nagasaki-u.ac.jp (M.M.); bb55319202@ms.nagasaki-u.ac.jp (M.O.); ozaki@nagasaki-u.ac.jp (Y.O.); ayoshi@nagasaki-u.ac.jp (A.Y.); 2Department of Regenerative Oral Surgery, Nagasaki University Graduate School of Biomedical Sciences, 1-7-1 Sakamoto, Nagasaki 852-8588, Japan; sohba@nagasaki-u.ac.jp; 3Department of Oral and Maxillofacial Surgery, Juntendo University Hospital, 3-1-3 Hongo, Bunkyo-ku, Tokyo 113-8431, Japan; asahina@nagasaki-u.ac.jp

**Keywords:** antimicrobial photodynamic therapy (a-PDT), periodontitis, periodontal probing depth (PPD), bleeding on probing (BOP), periodontal maintenance

## Abstract

Antimicrobial photodynamic therapy (a-PDT) in combination with scaling root planing (SRP) is more effective at improving periodontal status than SRP alone. However, the effectiveness of a-PDT in combination with irrigation for patients undergoing periodontal maintenance has not been clarified. This study evaluated the efficacy and safety of a-PDT in the maintenance phase. Patients who had multiple sites with bleeding on probing (BOP) and periodontal probing depth (PPD) of 4–6 mm in the maintenance phase were treated with a split-mouth design. These sites were randomly assigned to one of two groups: the a-PDT group and the irrigation group. In the a-PDT group, the periodontal pockets were treated with light-sensitive toluidine blue and a light irradiator. In the irrigation group, the periodontal pockets were simply irrigated using an ultrasonic scaler. After 7 days, the safety and efficacy of a-PDT were assessed. The mean PPD of the a-PDT group had reduced from 4.50 mm to 4.13 mm, whereas negligible change was observed in the irrigation group. BOP significantly improved from 100% to 33% in the PDT group, whereas it hardly changed in the irrigation group. No adverse events were observed in any patients. a-PDT may be useful as a noninvasive treatment in the maintenance phase, especially in patients with relatively deep periodontal pocket.

## 1. Introduction

Periodontitis is accompanied by inflammation of the periodontal tissue and destruction of alveolar bone, which may result in tooth loss [1]. The prevalence of periodontitis is estimated to be approximately 45% of adults in the United States [2]. The main cause of periodontal diseases is dental plaque, so initial periodontal treatment focuses on the removal of etiologic factors, such as dental plaque and calculus, by oral hygiene and scaling and root planing (SRP). For deep periodontal pockets, periodontal surgery may be necessary to accomplish thorough debridement [3]. The goal of periodontal treatment is to reach a clinically healthy state in which periodontal probing depth (PPD) is reduced to 3 mm or less and bleeding on probing (BOP) is not detected. However, successful treatment cannot be accomplished for all patients. In these cases, the inflammation of periodontal tissue is improved, but PPD and BOP are not reduced to healthy levels. Remission/control is an alternate goal for the treatment of these patients [4].

Maintenance is essential for the stabilization of diseases in patients being treated to the level of remission/control [5]. Residual periodontal pockets should be treated carefully during the maintenance phase. However, the frequency of SRP should be limited, because repeated SRP may cause damage to root surfaces. It is known that antimicrobial agents are also useful for treatment at these sites [6]. Antimicrobial photodynamic therapy (a-PDT) is one such method. In a-PDT, irradiation of a light-sensitive substance causes the production of reactive oxygen species, which have a bactericidal effect [7,8]. Unlike human tissue, bacteria and other microorganisms are greatly impacted by a-PDT absorption. Compared to other therapies, a-PDT is less invasive and can be used for almost all patients, other than those with photosensitivity. It can be applied more frequently than antimicrobials because a-PDT does not cause bacterial resistance. Research also suggests that a-PDT can inactivate endotoxins produced by Gram-negative bacteria [9,10] and has an effect on wound healing [11].

Many randomized controlled clinical trials have shown that a-PDT and SRP together cause significant improvements in PPD and clinical attachment level (CAL) compared to SRP alone [12,13,14,15,16]. Since a-PDT alone is less effective in eliminating biofilm than SRP [17], a-PDT should be used in combination with SRP in active periodontal treatments. On the other hand, evidence of the effectiveness of a-PDT in the periodontal maintenance phase is scarce. One study demonstrated that repeated a-PDT adjunctive with ultrasonic debridement was found to be more effective than repeated ultrasonic debridement alone in improvement in PPD, but not BOP, in patients undergoing maintenance [18]. However, no studies have compared the effectiveness of single a-PDT and irrigation combination therapy with that of irrigation alone in maintenance patients.

In the present study, patients who had multiple BOP-positive sites with PPD of 4–6 mm were treated with a split-mouth design. In the test sites, periodontal pockets were treated with the photosensitive substance toluidine blue and light irradiation in combination with irrigation using an ultrasonic scaler. In the control sites, the periodontal pockets were simply irrigated using the ultrasonic scaler. After one week, we compared the changes in PPD, BOP, plaque retention, and tooth mobility of the test and control sites and evaluated the safety of a-PDT.

## 2. Results

### 2.1. Effect of a-PDT on PPD

Each patient contributed two BOP-positive sites with PPD of 4–6 mm (split-mouth design), which were randomly assigned to one of the two treatments: irrigation and a-PDT (Figure 1).

There was no significant difference between the PPD of the irrigation group (*n* = 30) and that of the a-PDT group (*n* = 30) at baseline (Table 1). The PPD of the a-PDT group had significantly reduced 7 days after the treatment (*p* < 0.01), whereas the PPD of the irrigation group hardly changed (Figure 2A). The PPD of the irrigation group was significantly deeper than that of the a-PDT group 7 days after treatment (*p* < 0.01). Then, the irrigation and a-PDT groups were divided into two subgroups according to their PPD at baseline (4 mm and 5–6 mm). The PPD of each subgroup was not significantly different between the irrigation and a-PDT groups at baseline (Table 1). In the 4 mm PPD subgroup, PPD had not significantly changed in either the a-PDT or irrigation group (Figure 2B). In the 5–6 mm PPD subgroup, PPD had significantly reduced in the a-PDT group, but not in the irrigation group (Figure 2C). PPD in the irrigation group was significantly deeper than that in the a-PDT group 7 days after treatment (*p* < 0.01).

### 2.2. Effect of a-PDT on BOP

BOP was 100% positive in both the irrigation and a-PDT groups at baseline (Table 2). Seven days after treatment, the frequency of BOP was significantly different between the irrigation group (97%) and the a-PDT group (33%). Then, the irrigation and a-PDT groups were divided into two subgroups according to their PPD at baseline. In the 4 mm PPD subgroup, the frequency of BOP was not significantly different between the irrigation group (94%) and the a-PDT group (43%). In the 5–6 mm PPD subgroup, the frequency of BOP in the a-PDT group (8%) was substantially lower than that in the irrigation group (100%).

### 2.3. Effects of a-PDT on Plaque Retention and Tooth Mobility

Dental plaque was detected in 30 out of 30 sites in the irrigation group and 23 out of 30 sites in the a-PDT group at baseline (Table 3). Seven days after treatment, the number of plaque-detected sites was not different between the irrigation group (15 sites) and the a-PDT group (15 sites). In the 4 mm PPD subgroup, the number of plaque-detected sites was not significantly different between the irrigation group (13 sites) and the a-PDT group (9 sites). In the 5–6 mm PPD subgroup, the number of plaque-detected sites in the a-PDT group (6 sites) was significantly larger than in the irrigation group (2 sites). 

There was no significant difference between the tooth mobility of the irrigation group (0.57 ± 0.50) and the a-PDT group (0.60 ± 0.50) at baseline (Table 1). Tooth mobility did not change in either the irrigation group or a-PDT group 7 days after treatment, suggesting that a-PDT does not affect tooth mobility.

### 2.4. Adverse Events after a-PDT Treatment

No patient complained of general symptoms, such as fever, fatigue, or lack of appetite, after a-PDT treatment. No local adverse events, such as gingival swelling, spontaneous bleeding, or continuous pain, were observed at a-PDT-treated sites.

## 3. Discussion

The present split-mouth randomized controlled trial showed that a-PDT treatment for residual 4–6 mm periodontal pockets in the maintenance phase resulted in a significant improvement in PPD and BOP. The number of sites where plaque was detected was not significantly different between the a-PDT group and the irrigation group 7 days after treatment, suggesting that improvements in PPD and BOP were not due to differences in oral hygiene. Further analysis dividing these cases into two subgroups according to the PPD at baseline showed that a-PDT treatment was more effective in the 5–6 mm PPD subgroup than in the 4 mm PPD subgroup. Moreover, no local adverse events were observed at a-PDT-treated sites. These results suggest that a-PDT is useful for the treatment of residual periodontal pockets, especially in pockets with 5–6 mm PPD in the maintenance phase.

The use of a-PDT together with irrigation resulted in an apparent improvement in inflammation of periodontal tissue judging from the reduction of BOP. This was not due to a reduction in supragingival biofilm, because oral hygiene at the a-PDT group sites was no better than that of the irrigation group. Biofilm in deep periodontal pockets mostly consists of anaerobic Gram-negative bacteria [19]. a-PDT has been reported to be effective against aerobic and anaerobic bacteria, as well as Gram-positive and Gram-negative bacteria [20,21,22,23,24,25]. This bactericidal effect has been shown not only in bacteria but also in bacterial biofilm [26,27]. These effects of a-PDT may have contributed to the elimination of the anaerobic bacteria and bacterial biofilm that resisted irrigation in the maintenance treatment. This may be one of the reasons that a-PDT was more effective in improving BOP in pockets with 5–6 mm PPD than in pockets with 4 mm PPD. Furthermore, a-PDT has been shown to inhibit the bioactivity of lipopolysaccharides, which are bacterial toxins from Gram-negative bacteria [28]. Moreover, a-PDT can promote wound healing [11]. These additional effects of a-PDT may have also contributed to the improvement of inflammation in periodontal tissue.

Andersen et al. evaluated the effectiveness of a-PDT and SRP together in moderate to severe periodontitis patients, compared to SRP alone. In contrast to SRP alone, a-PDT and SRP together led to significant improvements in PPD and CAL at 6 and 12 weeks after treatment [12]. After this, numerous similar randomized controlled clinical trials were carried out, and accumulating evidence shows that a-PDT together with SRP leads to significant improvements in PPD and CAL over the use of SRP alone [12,13,14,15]. In these clinical studies, a-PDT was used in combination with SRP because a-PDT alone is less effective in eliminating biofilm than SRP [17]. We applied a-PDT combined with irrigation in patients undergoing maintenance because detectable biofilm and risk factors had already been removed from root surfaces. As a result, a-PDT combined with irrigation led to significant improvements in PPD and BOP over the use of irrigation alone. The amount of bacteria in the periodontal pockets of these patients were likely small enough to be eliminated by a-PDT. Muller Campanile et al. showed that a-PDT together with ultrasonic debridement repeated once or twice a week was more effective than ultrasonic debridement alone repeated once a week in improving PPD, but not in improving BOP, in the maintenance phase [18]. Irrigation combined with a-PDT was effective in our study, but ultrasonic debridement combined with a-PDT was not effective in improving BOP in their study. The effect of undergoing ultrasonic debridement once or twice a week was likely so effective that the adjunctive effect of a-PDT in improving BOP was not detected in their study.

The results of the present study demonstrated the usefulness of a-PDT in the maintenance phase. However, the effects of a-PDT were evaluated 7 days after treatment, and the effects at a later stage remain unknown. Systematic reviews have shown that a-PDT combined with SRP is effective in improving PPD and BOP when evaluated after short intervals [16,29]. On the other hand, a-PDT combined with SRP seems to be no longer effective after 3 months [30,31,32]. The effect of a-PDT in combination with irrigation should be evaluated at longer intervals in future studies. Since a-PDT does not cause bacterial resistance, the effects of repeated applications of a-PDT should be examined. Although only nonsmokers without severe systemic diseases were included in the present study, it has been reported that the effect of a-PDT is attenuated in smokers [32]. Several meta-analyses have shown that a-PDT combined with SRP in diabetic patients had no adjunctive effect over SRP alone [33,34]. Invasive treatments should be avoided in patients with severe systemic diseases, so a-PDT may be particularly useful for these patients. Thus, the effectiveness of a-PDT on these patients should be evaluated in future study. These studies will reveal the long-term efficacy and safety of a-PDT for patients undergoing periodontal maintenance with or without medical problems.

## 4. Materials and Methods

### 4.1. Study Population

Thirty patients who visited Nagasaki University Hospital for periodontal maintenance from November 2019 to April 2020 were included in this study. These patients had been treated with initial periodontal therapy and, if necessary, periodontal surgery prior to maintenance. All patients consented to the purpose of this study.

Patients who met all of the following criteria were included in the study:(1)Patients whose periodontal examination had revealed multiple BOP-positive sites with PPD of 4–6 mm.(2)Patients aged 20 years or older and less than 90 years old.(3)Patients who provided written informed consent to participate in this study of their own accord.

Patients with any of the following conditions were excluded from the study:(1)Patients lacking the ability to make decisions.(2)Photosensitive patients.(3)Patients hypersensitive to drugs, such as toluidine blue.(4)Patients taking antimicrobial or anti-inflammatory drugs.(5)Pregnant or lactating women.(6)Patients with severe diabetes, hepatic disease, renal disease, or other conditions leaving them particularly susceptible to infections.(7)Patients who were otherwise considered unsuitable for this study.

BOP-positive sites with PPD of 4–6 mm were selected for this study because active periodontal therapies, such as SRP and periodontal surgery, are usually necessary for the treatment of sites with PPD > 6 mm. The sites that met the selection criteria were treated with a split-mouth design, in which the a-PDT group and the irrigation group were randomized in a 1:1 ratio (Figure 1). The basic characteristics of the study population are shown in Table 1.

### 4.2. Ultrasonic Irrigation and a-PDT Treatment

In the irrigation group, the periodontal pockets were irrigated using an ultrasonic scaler (Solfy, Morita, Tokyo, Japan) at a power setting of 0.5 for 30 s (Figure 1). In the a-PDT group, the periodontal pockets were irrigated using the ultrasonic scaler at a power setting of 0.5 for 10 s, injected with a light-sensitive toluidine blue gel (0.1 mg/mL, FotoSan Agent, CMS Dental, Jutland, Denmark), and irradiated with a light-emitting diode (LED) irradiator (FotoSan 630; peak wavelength of 630 nm, output power 2000 to 4000 mW/cm^2^) using a long perio tip inserted into the pockets for 30 s. Then, the pockets were irrigated using the ultrasonic scaler for 10 s, injected with the light-sensitive substance again, irradiated from the buccal or lingual side for 10 s, and irrigated using an ultrasonic scaler for 10 s.

### 4.3. Evaluation of Periodontal Status

After the ultrasonic irrigation or a-PDT treatment, every patient maintained their own routine brushing and oral care during the course of the study. Their periodontal status was evaluated 7 ± 3 days after the treatment because the adjunctive effects of a-PDT to SRP or ultrasonic debridement were reported to last for at least 7 days [18,29]. The presence or absence of dental plaque at the treatment sites was evaluated after staining with Red-Cote (Sunstar, Tokyo, Japan). Tooth mobility was evaluated according to Miller’s mobility index. Two calibrated examiners measured PPD, BOP, plaque retention, and tooth mobility at the treated sites.

### 4.4. General and Local Complications

General complications, such as fever, fatigue, rash, itchiness, nausea, diarrhea, and lack of appetite, were assessed 7 ± 3 days after treatment. Local inflammatory symptoms, such as redness, swelling, pain, and ulcers, were also assessed 7 ± 3 days after treatment.

### 4.5. Statistical Analysis

Comparisons of PPD were carried out using paired or unpaired *t*-tests, while changes in BOP and dental plaques were analyzed using chi-square tests. A *p*-value < 0.05 was considered statistically significant. All calculations were conducted using StatMate V software (ATMS, Tokyo, Japan).

## 5. Conclusions

a-PDT combined with irrigation was effective in improving PPD and BOP in patients undergoing maintenance. No adverse events were observed at a-PDT-treated sites. Thus, a-PDT may be useful as a noninvasive treatment in the maintenance phase.

## Figures and Tables

**Figure 1 pharmaceuticals-15-00924-f001:**
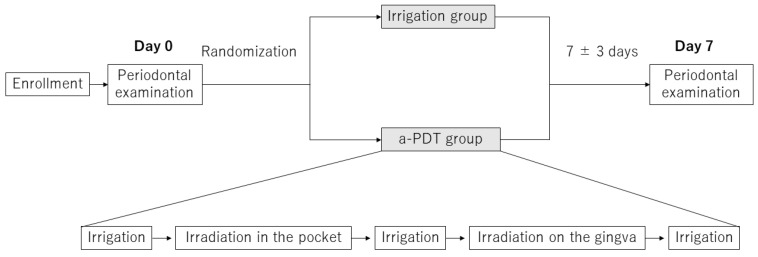
Study design. An ultrasonic scaler was used for irrigation.

**Figure 2 pharmaceuticals-15-00924-f002:**
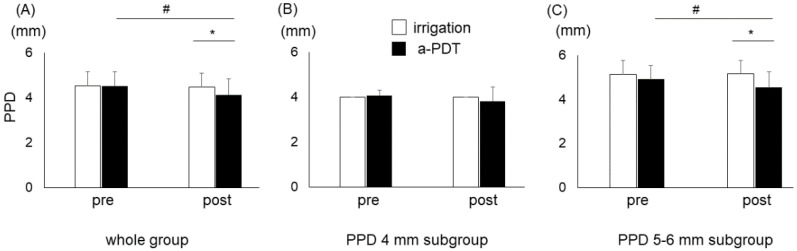
Periodontal probing depth (PPD) before and after treatment. PPD was measured at baseline and 7 days after irrigation or antimicrobial photodynamic therapy (a-PDT). Graphs show mean pre- and posttreatment PPD levels and standard deviations of the whole group (**A**), the 4 mm PPD subgroup (**B**), and the 5–6 mm PPD subgroup (**C**). # *p* < 0.01 in relation to comparisons between the treatments using paired *t*-test; * *p* < 0.01 in relation to comparisons between the groups using *t*-test.

**Table 1 pharmaceuticals-15-00924-t001:** Basic characteristics of the study population at baseline.

		Irrigation			a-PDT	
	Whole	4 mm	5–6 mm	Whole	4 mm	5–6 mm
n	30	16	14	30	17	13
Gender (M/F)	13/17	6/10	7/7	13/17	6/11	7/6
Age (years)	62.0 ± 14.2	59.5 ± 15.4	66.0 ± 11.1	62.0 ± 14.2	57.5 ± 15.5	67.8 ± 9.1
PPD (mm)	4.53 ± 0.62	4.00 ± 0.00	5.14 ± 0.36	4.50 ± 0.63	4.00 ± 0.00	5.07 ± 0.47
BOP (%)	100	100	100	100	100	100
Plaque (%)	100	100	100	77	100	46
Mobility	0.57 ± 0.50	0.44 ±0.51	0.71 ±0.47	0.60 ± 0.50	0.47 ± 0.51	0.77 ± 0.43
Smoking (%)	0	0	0	0	0	0

PPD, periodontal probing depth; BOP, bleeding on probing.

**Table 2 pharmaceuticals-15-00924-t002:** Number of BOP-positive sites before and after treatment.

		Irrigation	a-PDT	Odds Ratio	*p* Value *
Whole group	Pre	30 (100%)	30 (100%)	0.35	0.011
	Post	29 (97%)	10 (33%)		
4 mm PPD	Pre	16 (100%)	17 (100%)	0.57	0.42
	Post	15 (94%)	9 (53%)		
5–6 mm PPD	Pre	14 (100%)	13 (100%)	0.077	0.006
	Post	14 (100%)	1 (8%)		

BOP, bleeding on probing; a-PDT, antimicrobial photodynamic therapy; PPD, periodontal probing depth. * Comparison between the treatments using chi-square tests.

**Table 3 pharmaceuticals-15-00924-t003:** Number of plaque retention sites before and after treatment.

		Irrigation	a-PDT	Odds Ratio	*p* Value *
Whole group	Pre	30 (100%)	23 (77%)	1.30	0.65
	Post	15 (50%)	15 (50%)		
4 mm PPD	Pre	16 (100%)	17 (100%)	0.65	0.58
	Post	13 (81%)	9 (53%)		
5–6 mm PPD	Pre	14 (100%)	6 (46%)	7.00	0.044
	Post	2 (14%)	6 (46%)		

a-PDT, antimicrobial photodynamic therapy; PPD, periodontal probing depth. * Comparison between the treatments using chi-square tests.

## Data Availability

Data is contained within the article.

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
