# Peer review of "Clinical Efficacy and Safety of Antimicrobial Photodynamic Therapy in Residual Periodontal Pockets during the Maintenance Phase"

_pharmaceuticals, 2022, doi:10.3390/ph15080924_

Round 1

Reviewer 1 Report

Title           OK

 Abstract          OK

 Introduction

-          Page 2, line 59-76: whole paragraph

This entire paragraph does not correspond to the introductory part of the manuscript. Fits the discussion.

Also, the goal of this research is not clearly presented in the introductory part.

 Material and methods

Why is the Materials and Methods section placed after the Results and Discussion sections? 

 Results

In the Results section, the data that fit in the material and methods section were mixed with the obtained results. Therefore, it is necessary to separate these two parts.

Discussion

In the discussion section, there is no comparison of the results obtained from this research with the results of other authors.

The discussion must be reformulated and reworked.

 Conclusion

What is the conclusion of the research?

 General opinion:

The author chose a very interesting and today attractive issue for investigation.

In general, the manuscript is well thought out, the tests well done, but the results and discussion are not well presented and explained.

Their correction, division and systematization of data is necessary.

Author Response

 Thank you for giving us the important comments and suggestion.

We have addressed your comments as follows:

  1. Introduction, Page 2, line 59-76: whole paragraph. This entire paragraph does not correspond to the introductory part of the manuscript. Fits the discussion. Also, the goal of this research is not clearly presented in the introductory part.

According to the suggestion, we have rewritten the third paragraph of the Introduction. We have rewritten the last sentence of the Introduction to clarify the goal of this research.

  1. Material and Methods. Why is the Materials and Methods section placed after the Results and Discussion sections?

According to the ‘Instructions for Authors’ of Pharmaceuticals, the manuscript sections should appear in the following order: Introduction, Results, Discussion, Materials and Methods, Conclusions (optional). We have followed this guideline.

  1. Results. In the Results section, the data that fit in the material and methods section were mixed with the obtained results. Therefore, it is necessary to separate these two parts.

As written above, the Materials and Methods appears after Discussion. Therefore, some experimental procedure was described in the Results.

  1. Discussion. In the discussion section, there is no comparison of the results obtained from this research with the results of other authors. The discussion must be reformulated and reworked.

We have added the comparison of the results obtained from this research with the results of other authors to the Discussion.

  1. Conclusion. What is the conclusion of the research?

We have added the ‘Conclusion’ to section 5 (after the Materials and Methods).

Reviewer 2 Report

The manuscript sounds interesting, and the work is well written and well planned. Nevetheless, it seems so weird that Materials and Methods section has been placed at the end of the work. Introduction should be improved with a deep management of literature knowledge. 

Author Response

Thank you for giving us the important comments and suggestion.

We have addressed your comments as follows:

  1. It seems so weird that Materials and Methods section has been placed at the end of the work.

According to the ‘Instructions for Authors’ of Pharmaceuticals, the manuscript sections should appear in the following order: Introduction, Results, Discussion, Materials and Methods, Conclusions (optional). We have followed this guideline.

  1. Introduction should be improved with a deep management of literature knowledge.

According to the suggestion, we have rewritten the Introduction.

Reviewer 3 Report

The paper is interesting and in my opinion it could be published in special issue "Photodynamic Therapy 2022" of "Pharmaceuticals" after minor revisions regarding some issues presented below. 

It is not clear if Nagasaki University Hospital Certified Review Board has competence of Ethical Commission - if not, the number of the approval of the Ethical Commission for this trial should be presented.

In "Material and Methods" in subchapter "Ultrasonic irrigation and a-PDT treatment" the value of energy density of radiation emitted by light-emitting diode, of power of ultrasonic wave, as well as characteristics of toluidine blue (form, concentration etc.)  applied as photosenthitizer are lacking - it should be corrected.

The applied paired t-test is properly selected for comparison of the obtained data within particular groups (before and after the end of treatment), but they should not be used for comparison of the data between irrigation group and a-PDT group, both before and after the end of treatment. The statistical analysis with the use of proper tests (non-paired) should be performed in those cases.

In caption of Table 3 instead of the statement "Number of BOP-positive sites..." the statement "Number of dental plaque sites" should be used.

In my opinion a short subchapter stating the "Conclusions" should be placed after the "Discussion", that could enable the readers better interpretation of the obtained results.

Taking into account that statistically significant improvement after a-PDT treatment was observed only in 5-6 mm PPD group and not in 4 mm PPD group the conclusions, both in the text of the manuscript and in the "Abstract" should be more precise, for example: "a-PDT may be useful as a non-invasive treatment in the maintenance phase, especially in patients with relatively high periodontal probing depths".

In reference 17 the first and last page of the paper is lacking.

Author Response

Thank you for giving us the important comments and suggestion.

We have addressed your comments as follows:

  1. It is not clear if Nagasaki University Hospital Certified Review Board has competence of Ethical Commission - if not, the number of the approval of the Ethical Commission for this trial should be presented.

Nagasaki University Hospital Certified Review Board has competence of Ethical Commission. The number of the approval of the Ethical Commission is CRB19-013 and registered in the Japan Registry of Clinical Trials database (jRCTs072190034). This information is described in Page7, Line 302-304.

  1. In "Material and Methods" in subchapter "Ultrasonic irrigation and a-PDT treatment" the value of energy density of radiation emitted by light-emitting diode, of power of ultrasonic wave, as well as characteristics of toluidine blue (form, concentration etc.) applied as photosenthitizer are lacking - it should be corrected.

According to the suggestion, we have added the information on the value of energy density of radiation emitted by light-emitting diode, of power of ultrasonic wave, as well as characteristics of toluidine blue to the Materials and Methods.

  1. The applied paired t-test is properly selected for comparison of the obtained data within particular groups (before and after the end of treatment), but they should not be used for comparison of the data between irrigation group and a-PDT group, both before and after the end of treatment. The statistical analysis with the use of proper tests (non-paired) should be performed in those cases.

We are sorry for the confusion. The comparison of the data before and after the treatment was performed using paired t-test and comparison of the data between irrigation group and a-PDT group was performed using non-paired t-test. We have corrected the description in the legend of Figure 1 and the Statistical Analysis in Materials and Methods.

  1. In caption of Table 3 instead of the statement "Number of BOP-positive sites..." the statement "Number of dental plaque sites" should be used.

We are sorry for this mislabeling. We have corrected it.

  1. In my opinion a short subchapter stating the "Conclusions" should be placed after the "Discussion", that could enable the readers better interpretation of the obtained results.

We have added the ‘Conclusions’ after the Materials and Methods according to the Instruction for Authors.

  1. Taking into account that statistically significant improvement after a-PDT treatment was observed only in 5-6 mm PPD group and not in 4 mm PPD group the conclusions, both in the text of the manuscript and in the "Abstract" should be more precise, for example: "a-PDT may be useful as a non-invasive treatment in the maintenance phase, especially in patients with relatively high periodontal probing depths".

According to the suggestion, we have changed the last sentence of the Abstract to "a-PDT may be useful as a non-invasive treatment in the maintenance phase, especially in patients with relatively high periodontal probing depths".

  1. In reference 17 the first and last page of the paper is lacking.

We are sorry for the lack of the page number. We have added it.